# The Extraction Using Deep Eutectic Solvents and Evaluation of Tea Saponin

**DOI:** 10.3390/biology13060438

**Published:** 2024-06-14

**Authors:** Jianjun Guo, Nanshan Zhao, Yaxin Zhao, Hao Jin, Guozhi Sun, Jing Yu, Haihua Zhang, Jianzhong Shao, Meilan Yu, Dongfeng Yang, Zongsuo Liang

**Affiliations:** 1School of Life Science and Medicine, Zhejiang Sci-Tech University, Hangzhou 310018, China; 2Engineering Research Center for Eco-Dyeing and Finishing of Textiles, Ministry of Education, Zhejiang Sci-Tech University, Hangzhou 310018, China; 3College of Horticulture, Hainan University, Haikou 570228, China; 4Shaoxing Academy of Biomedicine, Zhejiang Sci-Tech University, Shaoxing 312030, China

**Keywords:** deep eutectic solvent, tea saponin, *Camellia oleifera Abel*, antioxidant, antibacterial

## Abstract

**Simple Summary:**

In order to improve the comprehensive utilization of oil tea seed meal and increase the comprehensive economic benefits of the oil tea industry. In this paper, the low eutectic solvent was used as the extraction system, and the ultrasonic extraction method was combined with the enzymatic method to jointly extract the tea saponins. It was further purified with a large-pore adsorption resin to develop a set of environmentally friendly, energy-saving, and efficient tea saponin extraction and purification processes. On this basis, the tea saponins extracted from oil tea seed meal were further analyzed for antioxidant capacity and bacteriostatic activity. The results provided the scientific experimental basis and theoretical support for the efficient use of oil tea by-products.

**Abstract:**

Tea saponins have high surface-active and biological activities and are widely used in chemicals, food, pharmaceuticals, and pesticides. Tea saponins are usually extracted using ethanol or water, but both methods have their disadvantages, including a negative impact on the environment, high energy consumption, and low purity. In this study, we explored an effective process for extracting tea saponins from tea meal using deep eutectic solvents combined with ultrasonic extraction and enzymatic techniques. The experimental results showed that a high extraction efficiency of 20.93 ± 0.48% could be achieved in 20 min using an ultrasonic power of 40% and a binary DES consisting of betaine and ethylene glycol (with a molar ratio of 1:3) at a material–liquid ratio of 1:35 and that the purity of the tea saponins after purification by a large-pore adsorption resin reached 95.94%, which was higher than that of commercially available standard tea saponin samples. In addition, the extracted tea saponins were evaluated for their antioxidant and bacteriostatic activities using chemical and biological methods; the results showed that the tea saponins extracted using these methods possessed antioxidant properties and displayed significant antibacterial activity. Therefore, the present study developed a method for using deep eutectic solvents as an environmentally friendly technological solution for obtaining high-purity tea saponins from tea meal oil. This is expected to replace the current organic solvent and water extraction process and has great potential for industrial development and a number of possible applications.

## 1. Introduction

*Camellia oleifera Abel*. is one of the four major woody plants globally known for their production of edible oil. This abundant crop is cultivated and distributed on a large scale in China. The seeds of *Camellia oleifera Abel*. are particularly well-suited for oil extraction, yielding a clear and flavorful tea oil [1] renowned for being highly nutritious and resistant to deterioration and, thus, of superior quality. A significant amount of seed pomace is generated during oil extraction in the *Camellia oleifera Abel*. industry. This pomace contains valuable compounds, including tea saponins, tea seed polysaccharides, and tea polyphenols.

The seed pomace of *Camellia oleifera Abel*. is particularly rich in tea saponins, which are a type of saponin compound primarily composed of pentacyclic triterpenoids. Tea saponins have both surface-active and biological properties. Numerous studies have demonstrated the anti-inflammatory [2], antioxidant [3], antibacterial [4], anti-tumor [5], and gastroprotective properties [6] of tea saponins. Zhang et al. [7] found that tea saponins greatly decreased the severity of dermatitis and scratching behavior in atopic dermatitis mice. Tea saponins also restored skin barrier function and prevented the generation of inflammatory factors in mouse serum and skin tissues. Tea saponins appear to have antipruritic and anti-inflammatory properties, making them a viable natural resource for treating dermatitis. Zhang et al. [5] discovered that tea saponin might decrease the growth of human breast cancer cells and cause cell cycle arrest in the S phase. Furthermore, it will limit breast cancer cells’ invasive potential; tea saponins inhibit the development of both Gram-positive and Gram-negative bacteria. Tea saponin was discovered to modify the cell membrane’s permeability and structure, hence inhibiting mycelial development and reducing cell adhesion and aggregation. Additionally, tea saponins have the potential to regulate lipid metabolism and enhance the effects of oxidative damage when combined with aerobic exercise [8]. In terms of practical applications, tea saponins are utilized as foaming agents, detergents, surfactants [9,10,11], and other similar agents.

Green chemistry is becoming a growing concern with the environment’s deterioration and technology’s continued growth. “Green chemistry” aims to create chemical products and processes that minimize or eliminate the use and generation of harmful substances. Deep eutectic solvents (DESs) [12] have gained popularity in this context. These solvents, which have a lower melting point than any of their individual components, are composed of two or three compounds in a specific stoichiometric ratio of hydrogen bond acceptors to donors. The chemical structural formulae of several hydrogen bond donors and acceptors are shown in Figure 1. Deep eutectic solvents have emerged as promising green solvents for the extraction of bioactive compounds from plants due to their low cost, ease of preparation, degradability, and recyclability. For instance, Zhen [13] extracted flavonoids from *Ampelopsis grossedentata* leaves, and Zheng [14] successfully used betaine and glycerol to extract total phenol from fine corn bran. DESs have been employed to efficiently extract nucleic acids [15], proteins, peptides [16], and other biomolecules. Moreover, they have a protein-stabilizing effect, preventing protein denaturation [17]. By utilizing DES instead of organic solvents in the extraction process, researchers have typically obtained higher yields.

In previous studies, DES has rarely been used to extract tea saponins. Yu [18] extracted tea saponins with choline chloride and methylurea configured DES. Under optimal extraction conditions, the extraction rate of tea saponins reached 94.36 mg/g, which was 27% higher, and the extraction time was 50% shorter compared to ethanol extraction. Additionally, the tea saponins were not changed during DES extraction. Tang [19] achieved the highest extraction rate of tea saponins with a ternary deep eutectic solvent consisting of l-proline, glycerol, and sucrose. The maximum extraction yield was 23.22 ± 0.28% at the optimized extraction time, DES concentration, and liquid–solid ratio. It was also determined that the extracted saponins were not changed during processing.

On this basis, some other methods were added to break the cells and, thus, improve the extraction rate of tea saponins. Different methods can be used to break cells apart, which facilitates the release of their contents and boosts the production of the targeted products. There are two primary categories of cell disruption techniques: mechanical and non-mechanical. The mechanical methods include ultrasonication, high-pressure homogenization, and oscillatory bead impact fragmentation, while the non-mechanical methods include enzymatic digestion, chemical fragmentation, and freeze–thaw fragmentation. Ultrasound-assisted enzyme extraction (UAEE) is used in many fields, including the extraction of polysaccharides [20,21] and resistant starch [22], as well as the preparation of peptides [23].

However, its extraction efficiency is low due to the high viscosity of deep eutectic solvents. To address the limitations of conventional deep eutectic solvent extraction, this study employed a water-containing deep eutectic solvent in combination with enzymatic and ultrasonic methods for tea saponin extraction. Additionally, an orthogonal experimental method was used to determine the optimal extraction conditions. Cytotoxic and antibacterial assays were carried out to evaluate the biological activity of the extracted tea saponins. The findings of this study have the potential to enhance the efficiency of tea saponin extraction, simplify the purification process, and expand the application of DESs in extracting bioactive compounds from various plant materials.

## 2. Experimental Materials and Methods

### 2.1. Materials and Reagents

*Camellia oleifera* seed pomace was obtained from Tong’an Company in Xiamen, Fujian, China. Standard tea saponins were purchased from Shanghai Yuanye Bio-technology Co., Ltd. (Shanghai, China). All organic reagents and cellulase (10,000 U/g) were supplied by Shanghai Macklin Biochemical Co., Ltd. (Shanghai, China). A reactive oxygen species fluorescence assay kit was provided by Shanghai Biyuntian Biotechnology Co., Ltd. (Shanghai, China). A superoxide dismutase kit was sourced from Nanjing Jiancheng Bioengineering Research Institute (Nanjing, Jiangsu, China). The bacterial strains used include *Escherichia coli*, *Staphylococcus aureus*, *Bacillus subtilis*, *Diplococcus pneumoniae*, *Enterococcus faecalis*, and *Pseudomonas aeruginosa*. These strains come from the China General Microbiological Culture Collection Center (CGMCC). The 7701 cells and HepG2 cells used were sourced from cell lines preserved in our laboratory. DEME medium and fetal bovine serum were obtained from Gibco (Grand Island, NY, USA). We also used a multi-function pulverizer (Yujian, Zhejiang, China) electric-blast drying oven (BGZ-246, Boxun, Shanghai, China), electronic balance (Mettler-Toledo, Greifensee, Switzerland), ultrasonic cleaner (Boxun, Shanghai, China), multi-function enzyme labeler (Thermo Fisher Scientific, Waltham, MA, USA), constant-temperature water bath (Boxun, Shanghai, China), magnetic stirrer (Boxun, Shanghai, China), freezing centrifuge (Beckman, Miami, FA, USA), biological safety cabinets (Suzhou, China), fluorescence microscopes (Olympus, Tokyo, Japan), fume hoods (Suzhou, China), and pH meters (Mettler-Toledo, Greifensee, Switzerland).

### 2.2. Preparation of Samples and Deep Eutectic Solvents

The *Camellia oleifera* seed pomace was dried in an electric thermostat oven at 60 °C for more than 48 h. Subsequently, the pomace was finely ground using a grinder, sifted through a 60-mesh sieve, and then stored in a cool area for further drying. 

The hydrogen bond acceptors (HBAs) and hydrogen bond donors (HBDs) were mixed in a beaker at a specific molar ratio. The mixture was heated in a water bath at 80 °C until a uniform and transparent liquid was obtained. The compositions and molar ratios of the DESs are shown in Table 1.

### 2.3. Quantitative of Tea Saponin

Tea saponin quantification was performed using the vanillin-sulfuric acid method [24]. Briefly, 20 mg of standard tea saponins was dissolved in a 20 mL volumetric flask with 80% ethanol to create a standard master batch at a concentration of 1 mg/mL. Various volumes of the standard master batch were then transferred to 10 mL centrifuge tubes, and their volumes were made up to 0.5 mL using distilled water. Subsequently, 5 mL of 77% concentrated sulfuric acid and 0.5 mL of 8% vanillin ethanol were added to the sample solution while it was on ice. The centrifuge tubes were then placed in a water bath at 60 °C for 20 min for color development. After cooling the tubes at 0 °C for 10 min, they were brought to room temperature. A spectrophotometer was used to measure their absorbance at 550 nm.

### 2.4. HPLC Analysis of Tea Saponin

Precisely 10 mg of standard tea saponins was dissolved in chromatographic-grade methanol to create a 1 mg/mL solution of standard tea saponins. This solution was further diluted with methanol to generate various concentrations for chromatographic analysis. These standard solutions were subjected to chromatographic analysis after being filtered through a 0.45 μm microporous filter membrane. The experimental setup utilized a C18 chromatographic column under the following conditions: a column temperature of 30 °C, a mobile phase composed of methanol and water with equal elution (35:65, *v*/*v*), a flow rate of 1.0 mL/min, a detection wavelength at 265 nm, and an injection volume of 10 μL. Chromatograms were recorded, and a linear regression analysis was conducted using the peak area of tea saponins as the dependent variable and the concentration of the injected mass as the independent variable. The regression equation is shown in the Appendix A. The contents of the tea saponins were subsequently calculated based on the sum of the peak areas of the tea saponin chromatograms.

### 2.5. Screen of Deep Eutectic Solvents

The best solvent was chosen by figuring out which ratio of DES had the maximum solubility of tea saponins, and this was used to accelerate the extraction rate of tea saponins. For this, 0.1 g of sample powder was added to 10 mL of water-containing DES and agitated overnight on a shaker. The mixture was then subjected to ultrasonication and centrifuged. The resulting suspension was used to quantify the tea saponin content extracted via the vanillin–sulfuric acid method.

### 2.6. Optimization of Extraction Process by an Orthogonal Method

The best DES screened in 2.5 was used as the solvent for the mechanical extraction, ultrasonic extraction, and enzymatic digestion of tea saponins. Orthogonal experiments were conducted at room temperature to optimize the extraction conditions. Firstly, single-factor experiments were performed to investigate the effects of ultrasonic extraction time (A, min), material-to-liquid ratio (B), and ultrasonic power (C, w) on the extraction rate. Subsequently, the best experimental conditions were selected based on the results of the single-factor experiments, and an orthogonal array of 3-3 was employed to examine the synergistic effect of these three factors on the extraction rate of tea saponins. Ultimately, the optimal conditions for tea saponin extraction were determined. To increase the extraction rate, cellulase was added to the system.

### 2.7. Purification of Tea Saponins from Camellia oleifera Abel Seed Pomace

The methodology employed by Tang [19], which involved using microporous resin, was utilized for the isolation and purification of tea saponins. Before adsorption onto D101 microporous resin, the resin underwent pre-soaking in 95% ethanol, 5% HCl, and 5% NaOH for 12 h, with neutralization carried out after each soaking. Then, three times the extracted solvent’s volume of 90% ethanol was added, and this was kept at 75 °C for 3 h. The resulting precipitate was removed, and the supernatant was collected and concentrated using a rotary evaporator. The concentrated solution was then subjected to adsorption using the pre-treated resin, which was subsequently eluted with deionized water and 80% ethanol. The eluate was collected and concentrated using a rotary evaporator. After two rounds of elution and concentration, the concentrate was lyophilized in a freezing dryer. The lyophilized tea saponin samples were used in all subsequent tests.

### 2.8. Evaluation of the Biological Activity of Tea Saponins

#### 2.8.1. Antioxidant Activity Assay

The antioxidant activity of DES-extracted tea saponins was assessed by evaluating their capacity to scavenge free radicals such as superoxide radicals [25] (O^2−^) and DPPH [26,27]. A DPPH solution was prepared by dissolving 0.0197 g of DPPH in anhydrous ethanol. To obtain different concentrations, the extracted tea saponin samples were dissolved in 70% ethanol using a twofold dilution method. Subsequently, 1 mL of the sample solution and 1 mL of the DPPH solution were added to a tube, with anhydrous ethanol used as a blank control. The mixture was then left to stand in the dark for 30 min, and its absorbance was measured at 517 nm using a spectrophotometer. 

The scavenging rate of superoxide radicals was determined using pyrogallol. A 0.4 mg/mL pyrogallol solution was prepared as follows: 0.02 mL of pyrogallol solution was mixed with 1 mL of PBS solution (pH = 8.0), 2 mL of sample solution was added, and the volume was adjusted to 2 mL. Its absorbance was measured at 320 nm immediately after a 5 min reaction. The results were then plotted into graphs according to Equation (1).
(1)$$Clearance\;rate=\frac{A_{0}-A_{1}}{A_{0}}$$
*A*_0_—Blank tube *A*_1_—Sample liquid tube.

#### 2.8.2. Cytotoxicity Assay

The cells used were HepG2 human hepatocellular carcinoma cells and 7701 human hepatocytes. The frozen cells were resuscitated and prepared as a cell suspension. Previous experiments have shown that both cell types can survive at concentrations ranging from 20 to 100 ng/μL. As such, three concentrations were selected for the present experiment: 20, 50, and 100 ng/μL. The tea saponin samples obtained in 2.7 were diluted to these concentrations. The cell was treated with diluted extracts for a duration of 3 h. At the same time, their cell morphology was observed and recorded using a fluorescence microscope. The impact of tea saponins on reactive oxygen species [28,29] levels was assessed using a reactive oxygen species fluorometric kit. The protein concentration in the cell extracts was quantified using the BCA kit. The SOD activity [30,31] in the cells was measured using the cellular superoxide dismutase kit. The SOD inhibition rate was calculated using Equation (2). SOD activity was calculated using Equation (3).
(2)$$SOD\;inhibition\;rate(\%)=\frac{{∆A}_{1}-{∆A}_{2}}{{∆A}_{1}}$$
(3)$$SOD\;Vitality\left(U/mgprot\right)=i÷50\%\times \frac{V_{1}}{V_{2}}\times f÷C_{pr}$$

Δ*A*_1_: the Control well OD value—the Control blank well OD value.Δ*A*_2_: the OD value of assay wells—the OD value of control blank wells.i: SOD inhibition rate (%).*V*_1_: the total volume of reaction solution (240 μL).*V*_2_: the volume of sample added (20 μL).f: the dilution multiple of the sample before adding it to the assay system.C_pr_: the protein concentration of the sample (mgprot/mL).

#### 2.8.3. Antimicrobial Activity Assay

Two experiments were conducted to confirm the inhibitory effect of the extracted tea saponins on the strains: inhibition circle experiments [32,33] and bacterial growth experiments [34]. The extracted tea saponin sample was dissolved in 70% ethanol to prepare a solution of 1.5 mg/mL, and this was then diluted. The concentration was 1.5 mg/mL for the high-concentration group, 0.75 mg/mL for the medium-concentration group, and 0.375 mg/mL for the low-concentration group. A total of six species were determined to be inhibited, including *Escherichia coli*, *Bacillus subtilis*, *Staphylococcus aureus*, *Enterococcus faecalis*, *S. pneumoniae* R, and *Pseudomonas aeruginosa*. The test strains were cultured in a medium until they reached the logarithmic growth phase. Then, the sample solution and bacterial solution were added to a centrifuge tube. The culture was maintained at a constant temperature of 37 °C, with continuous shaking at 150 rpm. At various time intervals, the absorbance of the suspension was measured at 600 nm. The resulting data were then plotted on a graph as the bacterial growth curve.

The Oxford cup method [35] was employed to conduct experiments on bacterial inhibition circles. Depending on the strain, different media configurations were made. After the medium solidified, the plates were coated with the desired bacterial solution, and the Oxford cups were placed on the medium, equally spaced. Subsequently, 100 μL of various dilutions of the extract were added to each Oxford cup. After a cultivation period, the diameter of the inhibition circle was measured. Ampicillin was used as the control group.

#### 2.8.4. Statistical Analysis

The results are expressed as the means ± SD. Significance analyses were performed using SPSS 25 software. Images were created using Graphpad Prism 8.

## 3. Results and Discussion

### 3.1. Synthesis of Deep Eutectic Solvents

The DESs were synthesized using a conventional heating method, combining three hydrogen bond donors (glycerol, urea, and ethylene glycol) with three hydrogen bond acceptors (choline chloride, betaine, and L-proline). Ten diverse DESs were successfully synthesized at various molar ratios, including six choline chloride-based DESs and four betaine-based DES (Table 2). All DESs demonstrated both transparency and stability at room temperature. Notably, during their preparation, it was observed that the solution containing L-proline as the hydrogen bond acceptor displayed heightened viscosity and inadequate fluidity. The betaine-based DESs tended to solidify at room temperature in the absence of water, impeding the establishment of stable solutions. In contrast, it was found that choline chloride, as a hygroscopic crystal, readily formed a solution during DES synthesis. Nevertheless, certain anhydrous choline chloride-based DES formulations displayed a propensity to freeze when subjected to refrigeration; only when there is an adequate moisture content can the flow dynamics be stably maintained. Thus, the utilization of aqueous DESs holds considerable promise for subsequent experimental investigations.

### 3.2. Preparation and Screening of Deep Eutectic Solvents

In this study, a solvent with a volume ratio of 3:7 of DES to water was employed for screening, compared to the 80% ethanol and water used as the control group. The experimental results are illustrated in Figure 2. It was observed that when using ChCl-based deep eutectic solvents, ChCl-based glycerol (ChCl-Gly) yielded a higher extraction rate compared to ChCl-based ethylene glycol (ChCl-EG). This can be attributed to the higher solubility of polar tea saponins in polar solvents, resulting in a higher tea saponin extraction rate. An increase in the glycerol molar ratio leads to a higher tea saponin extraction rate, as glycerol acts as the hydrogen bond donor in the solvent. Similarly, in solvents where ethylene glycol was the hydrogen bond donor, higher molar ratios of ethylene glycol correlated with increased extraction rates. The composition of deep eutectic solvents in their liquid state significantly influences their chemical properties. Incorporating additional hydroxyl or carboxyl groups into these solvents promotes the formation of more hydrogen bonds, thereby enhancing both the liquid’s stability [36] and extraction rates.

As seen in Figure 2, tea saponins were more soluble in the betaine-based deep eutectic solvent made from betaine and ethylene glycol. The effect of ethylene glycol’s molar share on the extraction rate was insignificant; meanwhile, a rise in the molar ratio of ethylene glycol caused a drop in viscosity.

Given its cost, extraction rate, viscosity, and experimental feasibility, a deep eutectic solvent consisting of betaine and ethylene glycol in a 1:3 molar ratio (Bet-EG (1:3)) was chosen as the optimal solvent for the extraction of tea saponins.

### 3.3. Optimization of Extraction Conditions

Various techniques can be used to extract tea saponins, such as heating, ultrasound-assisted, or microwave-assisted methods. In this study, tea saponins were primarily extracted from *Camellia oleifera* seed pomace using a DES in conjunction with ultrasound-assisted enzyme extraction. The extraction conditions were optimized using a three-factor, three-level orthogonal method, with betaine-glycol (1:3) DES as the extractant. The DES used contained 70% water to reduce its viscosity. The three considered factors were ultrasound power, ultrasound time, and the material-to-liquid ratio. Single-factor experiments were carried out for each of the three influencing factors individually.

The results of the single-factors experiment are shown in Appendix A. The results showed that an ultrasound power below 250 W increased the permeation, diffusion, and dissolution rates of tea saponin extraction. Nevertheless, excessive ultrasound power accelerated the breakdown of impurities in the Camellia oleifera seed pomace, such as proteins, polysaccharides, etc. This led to the formation of insoluble complexes containing tea saponins, hindering its extraction rate. A positive correlation was observed between the ultrasound’s duration and the tea saponin extraction rate. However, an overly extended duration led to a marked reduction in the tea saponin extraction rate. This was attributed to the enhanced leaching of water-soluble impurities, which enveloped the tea saponin molecules and impeded their diffusion into the solution. Furthermore, excessively extended ultrasound times result in increased energy consumption. The liquid-to-material ratio also influenced the tea saponin extraction rate: a low liquid-to-material ratio led to a slower penetration of the extraction solvent into the sample, reducing molecular diffusion and resulting in a decreased extraction rate; however, with an increasing liquid-to-material ratio, the quantity of extractant obtained also increased, leading to a greater dissolution of the contaminants. Furthermore, using additional solvent does not improve the extraction rate once the tea saponins’ dissolution reaches equilibrium; however, it results in excessive solvent consumption, increasing extraction costs and making subsequent recovery efforts more challenging.

Through an analysis of the single-factor experiments and the use of the orthogonal method, the factors were ranked in relation to their influence on extraction efficiency, with ultrasound power exerting the greatest influence and the liquid-to-material ratio the least. The optimal combination of extraction parameters was determined as A3B2C1, which corresponded to a liquid-to-material ratio of 35 to 1, a 20-min duration, and a 40% power level. The results of the orthogonal experiments are tabulated in Appendix A. Under these conditions, the experiment yielded a tea saponin extraction rate of 20.926% and a tea saponin concentration of 5.98 mg/mL.

### 3.4. Quantitative and Qualitative Determination of Tea Saponin Content

The vanillin–sulfuric acid method was used to determine that the equation of the standard curve was Y = 0.3014 ∗ X + 0.0096, with Y representing absorbance and X representing concentration. The R2 value of 0.9908 indicated that there was a strong correlation between absorbance and concentration.

The chemical composition of the extracted tea saponins was analyzed using the HPLC method. A photodiode array detector (PDA) was employed under the same chromatographic conditions. The absorption chromatography peak area was determined to be 265 nm, which corresponded to the maximum absorption peak. Figure 3a displays the chromatogram of the tea saponin standard at 265 nm, while Figure 3b presents the chromatogram of the extracted saponins. Subsequently, a regression equation (Y = 632,911 ∗ X + 25,875) was derived for the standard curve based on its peak area and mass concentration. Its linear regression coefficient (R2) was calculated to be 0.998, indicating a robust linear relationship between mass concentration and peak area within the range of 0.1 mg/mL to 1.0 mg/mL. Under identical chromatographic conditions, peaks with the same retention time as the tea saponin standard were observed in the sample, confirming the presence of tea saponins in the extract. A purity assessment of both the standard and the extract was conducted by analyzing the ratio of the peak areas of the two primary peaks to the total peak areas. The tea saponin standard product was labeled as 98% pure, but it was found to be 91.22% pure. However, the extract exhibited a purity of 95.94%, surpassing the standard’s purity.

Tea saponins were extracted under optimized experimental conditions and purified to obtain tea saponin samples. In the method established in this study, the extraction solvent can be recycled and reused, which reduces the resources wasted and is environmentally friendly.

### 3.5. Antioxidant Activity of Tea Saponin

The present study assessed the DPPH and O^2−^ scavenging capacities of tea saponins, with L-ascorbic acid used as the positive control. The results showed a positive correlation between the concentration of the extracted tea saponins and their antioxidant activity, although scavenging rates were lower than those of L-ascorbic acid. (Figure 4a,b). The DPPH clearance was 50.74% when the concentration of the tea saponin extract was 1.5 mg/mL. Superoxide anion radical scavenging was 35.59% when the concentration of the tea saponin extract was 1.5 mg/mL. Hu [37] obtained saponin extracts from oil tea cake using nbutanol and assayed their antioxidant activity; the IC50 for their DPPH radical scavenging was 3866 ± 3 μg/mL. Our results are consistent with Hu’s findings that tea saponins have antioxidant properties.

### 3.6. Cytotoxicity of Tea Saponin

This study investigated the effects of tea saponins on cellular superoxide dismutase (SOD) activity and reactive oxygen species (ROS) levels. SOD is a pivotal antioxidant enzyme responsible for eliminating free radicals. Its activity directly reflects an organism’s oxidative–antioxidant equilibrium and its ability to scavenge free radicals. Figure 5A indicated that the addition of 20 ng/μL of tea saponins increased the SOD activity in 7701 cells compared to the control group, while 50 ng/μL and 100 ng/μL of tea saponins exhibited inhibitory effects. This likely stems from the generation of superoxide anion radicals by 7701 cells in response to low tea saponin concentrations, leading to increased SOD activity. The inhibitory effect of 20 ng/μL of tea saponins on the SOD activity of HepG2 cells was not statistically significant. However, concentrations of 50 and 100 ng/μL of tea saponins significantly reduced the SOD activity of HepG2 cells. This may be attributed to excessive tea saponins inducing apoptosis in these cells, disrupting their antioxidant system and diminishing SOD activity. The observed phenomenon might be attributed to the anticancer properties of tea saponins in diverse types of cancer cells, supported by their ability to induce autophagy and apoptosis, as documented in prior studies [5,38,39]. Increasing the tea saponin concentration was associated with a significant decline in SOD viability and an elevated rate of cell mortality, demonstrating their clear dose-dependent relationships. 

Figure 5B illustrates the growth status of these cells under various tea saponin doses. Notably, untreated cells maintained their normal cellular architecture, while cells exposed to increasing tea saponin concentrations exhibited reduced cell counts and signs of damage. These findings suggest that tea saponins inhibit the growth of both cell types to some extent, while excessive doses may result in cell death.

ROS plays a vital role in cellular signaling and organismal homeostasis. Elevated ROS levels under stressful conditions can lead to oxidative stress, potentially damaging cellular structures. Oxidative stress is a well-established contributor to aging processes and various diseases. As shown in Figure 5C,D, varying concentrations of tea saponins were observed to reduce ROS levels in 7701 cells. Notably, the ROS level in HepG2 cells increased after a 20 ng/μL tea saponin treatment; however, with the continued increase in tea saponin concentration, the ROS levels in HepG2 cells decreased. This phenomenon can be attributed to low tea saponin concentrations damaging cancer cells, resulting in elevated ROS levels. As the tea saponin concentration rose, it induced cell death and a subsequent reduction in fluorescence intensity. 

Tea saponins have a damaging effect on cells, and especially on cancer cells. The detected decrease in fluorescence intensity is due to the cell death caused by tea saponins; when the saponin concentration is too high, it causes cell death, decreasing both the ROS level and the SOD activity of the cells.

### 3.7. Antibacterial Activity of Tea Saponin

This study explored the inhibitory effects of tea saponins on various microorganisms. The results indicate that 1.5 mg/mL of tea saponins exhibited a modest inhibition against a diverse array of microorganisms. Table 3 shows the positive correlation between tea saponin concentrations and their inhibitory effects. The various extract concentrations displayed weaker inhibitory effects on each strain compared to the positive control. Among the tested strains, tea saponins exhibited an inhibition closest to that of the positive control against Streptococcus pneumoniae R but showed its weakest inhibition against *S. aureus*. Bacillus subtilis displayed inhibition zones in response to all tea saponin concentrations, with no significant differences observed among the different concentration groups. The bacterial growth process, including the lag, log, stationary, and decline phases, can be observed in bacterial growth curves. Adding tea saponin extract to bacterial solutions delayed the onset of the logarithmic phase of each bacterium. This indicates that tea saponins decelerate bacterial growth. Additionally, during the stationary phase, the viable bacteria population was significantly smaller in the tea saponin-treated groups compared to the negative control. Tea saponins have antibacterial effects, and their antibacterial efficacy is significantly concentration-dependent (Figure 6). In summary, tea saponins have a growth-inhibiting effect on both Gram-positive and Gram-negative bacteria. This is shown by the fact that tea saponins inhibit the growth of bacteria in media and slow down the growth of bacteria.

## 4. Conclusions

The aims of this study were to explore the extraction process of tea saponins using a DES combined with enzymatic and ultrasound-assisted extraction and to compare the differences between the tea saponins extracted using DESs and conventional methods. The antioxidant and bacteriostatic activities of the tea saponins extracted using our proposed process were also evaluated. In conclusion, this study successfully developed an efficient process for obtaining high-purity tea saponins, the purity of which could easily reach 95.94%, which is higher than that of standard commercially available tea saponins. The prepared tea saponins possessed obvious antimicrobial and antioxidant properties. This study lays the foundation for the extraction of tea saponins using deep eutectic solvents, which have the potential to replace traditional alcoholic and aqueous extraction processes and are expected to be applied in the fields of food, cosmetics, and pharmaceuticals. However, further studies on the bioactivity of tea saponins are needed to fully exploit their medicinal potential.

## Figures and Tables

**Figure 1 biology-13-00438-f001:**
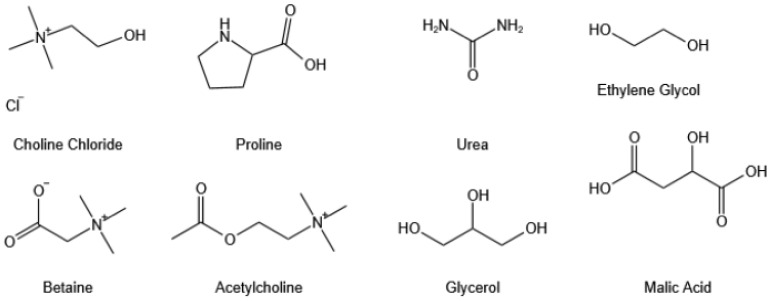
Chemical structure formula of partial hydrogen bond donor and hydrogen bond acceptor. Hydrogen bond accepter: Choline Chloride (ChCl), Betaine (Bet); Hydrogen bond dnor: Proline (Pro), Urea (UR), Ethylene Glycol (EG), Acetylcholine (Ach), Glycerol (Gly), Malic Acid (MA).

**Figure 2 biology-13-00438-f002:**
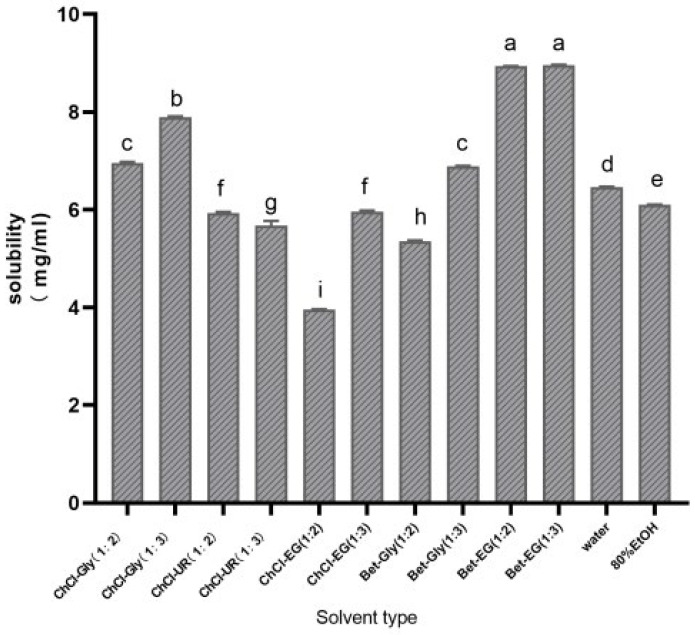
Screening of deep eutectic solvents for the extraction of tea saponins from Camellia oleifera seed pomace. The maximum solubility of tea saponins in different solvents was used to select the optimal solvent. Values marked with different letters indicate significant differences (*p* < 0.05).

**Figure 3 biology-13-00438-f003:**
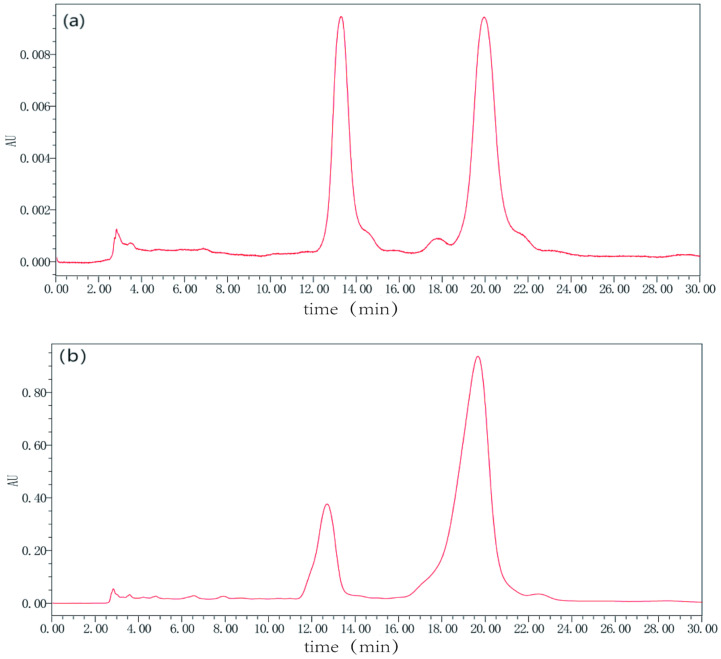
Liquid chromatography of tea saponin standard and extract. The mobile phase is methanol: water, with equal elution (35:65, *v*/*v*), a flow rate of 1.0 mL/min, detected the wavelength of 265 nm, and injection volume of 10 μL. The X-axis is real-time, and the Y-axis is the absorbance unit. The peak time of the two samples is the same. (**a**) is the liquid chromatogram of the standard, (**b**) is the liquid chromatogram of the extract.

**Figure 4 biology-13-00438-f004:**
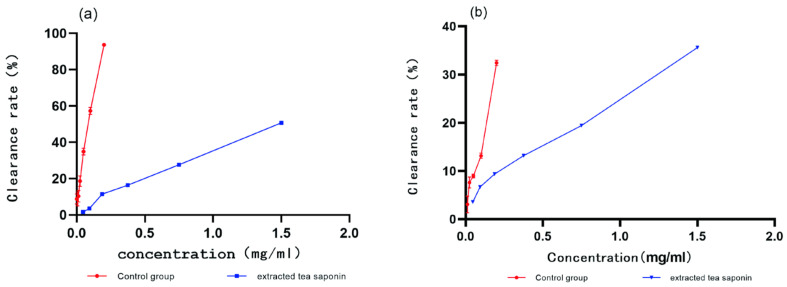
Evaluation of the antioxidant activity of tea saponins (**a**) DPPH clearance rate, with L-ascorbic acid as a positive control (**b**) superoxide free radical clearance rate, with L-ascorbic acid as a positive control.

**Figure 5 biology-13-00438-f005:**
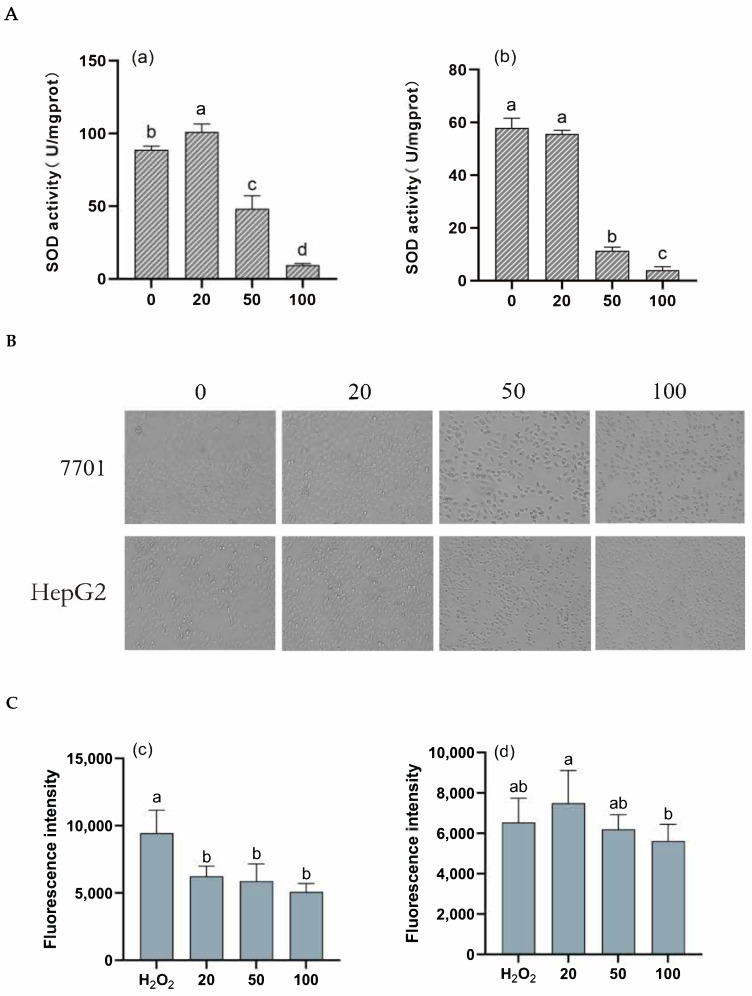
SOD activity and ROS levels of tea saponins in 7701 liver cells and HepG2. (**A**) SOD activity in 7701 (a), SOD activity in HepG2 (b): SOD activity was measured after 3 h of incubation with various concentrations of tea saponin in 7701 liver cells and HepG2. The values denoted with various letters indicate statistically significant differences (*p* < 0.05). With a rise in tea saponin content, SOD activity declines. (**B**) Morphology of 7701 and HepG2: treated cells with various tea saponin concentrations for three hours, then observe the morphology of the cells. The cell shape rapidly deteriorates, and the quantity of dead cells rises as tea saponin concentration rises. (**C**) ROS levels in 7701 (c), ROS levels in HepG2 (d) To evaluate ROS levels, incubate HepG2 and 7701 liver cells with different concentrations of tea saponin extract for 3 h. Significant differences are shown by values denoted by various letters (*p* < 0.05). Tea saponin concentration rises as the ROS level falls. (**D**) Image under fluorescent microscope: The overall fluorescence of 7701 was unsatisfactory, and when the concentration of tea saponin increased, the fluorescence became barely detectable. HepG2 fluorescence intensity decreased as tea saponin concentration increased. Scale bar = 500 μm.

**Figure 6 biology-13-00438-f006:**
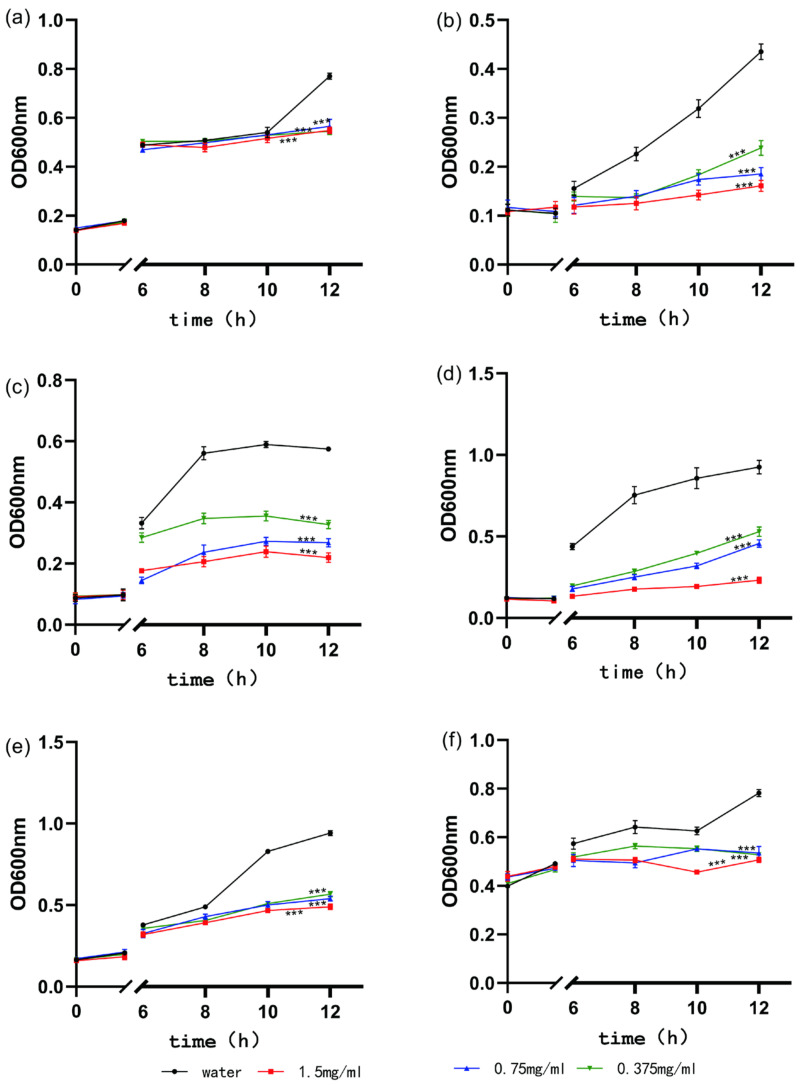
Effect of extracts on the growth curves of different strains of bacteria (**a**) *E. coli* (**b**) *Bacillus subtilis* (**c**) *S. pneumoniae R* (**d**) *Staphylococcus aureus* (**e**) *Enterococcus faecalis* (**f**) *Pseudomonas aeruginosa*, distilled water was used as a negative control. *** indicate significant differences, *p* < 0.01.

**Table 1 biology-13-00438-t001:** Components and mole ratios of the configured deep eutectic solvents.

Hydrogen Bond Acceptors	Hydrogen Bond Donors	Mole Ratio
ChCl	UR	1:2
	UR	1:3
	Gly	1:2
	Gly	1:3
	EG	1:2
	EG	1:3
Bet	UR	1:2
	UR	1:3
	Gly	1:2
	Gly	1:3
	EG	1:2
	EG	1:3
Pro	UR	1:2
	UR	1:3
	Gly	1:2
	Gly	1:3
	EG	1:2
	EG	1:3

**Table 2 biology-13-00438-t002:** Components and mole ratios of the successfully configured deep eutectic solvents.

Hydrogen Bond Acceptors	Hydrogen Bond Donors	Mole Ratio
ChCl	UR	1:2
	UR	1:3
	Gly	1:2
	Gly	1:3
	EG	1:2
	EG	1:3
Bet	Gly	1:2
	Gly	1:3
	EG	1:2
	EG	1:3

**Table 3 biology-13-00438-t003:** Antibacterial zone experiments of tea saponins on various bacteria, with results expressed as mean ± standard deviation and values marked with different letters indicating significant differences (*p* < 0.05), Ampicillin was used as positive control.

Strain	Positive Control (mm)	1.5 mg/mL (mm)	0.75 mg/mL (mm)	0.375 mg/mL (mm)
*E. coli*	46.377 ± 0.52 a	12.023 ± 0.44 b	11.060 ± 0.28 c	9.006 ± 0.12 d
*Bacillus subtilis*	49.101 ± 0.17 a	11.021 ± 0.20 b	11.139 ± 0.54 b	10.134 ± 0.37 c
*S. pneumoniae R*	32.037 ± 0.22 a	9.010 ± 0.20 b	8.079 ± 0.08 c	6.004 ± 0.09 d
*Staphylococcus aureus*	59.209 ± 0.19 a	13.008 ± 0.19 b	12.031 ± 0.10 c	10.040 ± 0.12 d
*Enterococcus faecalis*	47.019 ± 0.11 a	15.014 ± 0.10 b	12.080 ± 0.14 c	11.010 ± 0.05 d
*Pseudomonas aeruginosa*	30.023 ± 0.13 a	14.060 ± 0.11 b	11.054 ± 0.13 c	9.037 ± 0.12 d

## Data Availability

Data will be made available on request.

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
