# Peer review of "The Extraction Using Deep Eutectic Solvents and Evaluation of Tea Saponin"

_biology, 2024, doi:10.3390/biology13060438_

Round 1
Reviewer 1 Report
Comments and Suggestions for Authors
Dear Editor of journal of biology,
The manuscript titled "The extraction and evaluation of tea saponin using deep eutectic solvents". The idea of the manuscript is not new and it does not contain a kind of novelty. But still the work is good. So, I have some comments on this work which if the author takes all of them the manuscript will be improved.
Title" The extraction and evaluation of using deep eutectic solvents" could be changed to be
"Tea saponin extraction and evolution using deep eutectic solvents"
Abstract: the majority of the abstract is just method and the manuscript it will not published as method, so, please divided the abstract into three main regions as we always did, introduction , materials and methods, results and discussion. Additionally, ad the net conclusion and a convey resulted from this work.
Introduction:
Yes the paper carried out on the extraction of sponines from tea and authors wite on the importance of the tea and the types of the methods used in saponines extraction but they did not mentioned the most important thing of this work "Sponines". What is the importance of sponines, the work will not important if we did this work in non important materials. Authors should write a paragraph on the importance of sponines.
Results and discussion
Figure 4 and b should be collected in one figure as in figure 5.
Figure 6 is not clear please make the contrast is more good.
Citation is very low, please ad more citation and discuss your results based on the others results.
Conclusion
Is too long and should be shorten to half.
Author Response
|
1. Summary |
|
|
|
Thank you very much for taking the time to review this manuscript. Please find the detailed responses below and the corresponding revisions/corrections highlighted/in track changes in the re-submitted files. |
||
|
2. Point-by-point response to Comments and Suggestions for Authors |
||
|
Comments 1: Title" The extraction and evaluation of using deep eutectic solvents" could be changed to be"Tea saponin extraction and evolution using deep eutectic solvents" |
||
|
Response 1: Thank you for pointing this out. We agree with this comment. Therefore, we have made cuts and changes to the summary section
|
||
|
Comments 2: Abstract: the majority of the abstract is just method and the manuscript it will not published as method, so, please divided the abstract into three main regions as we always did, introduction , materials and methods, results and discussion. Additionally, ad the net conclusion and a convey resulted from this work. |
||
|
Response 2: Thank you for pointing this out. We agree with this comment. Therefore, we have made cuts and changes to the summary section
|
||
|
Comments 3: Yes the paper carried out on the extraction of sponines from tea and authors wite on the importance of the tea and the types of the methods used in saponines extraction but they did not mentioned the most important thing of this work "Sponines". What is the importance of sponines, the work will not important if we did this work in non important materials. Authors should write a paragraph on the importance of sponines. |
||
|
Response 3: Agree,We have added a discussion of the properties and applications of tea saponins in the introductory section.
|
||
|
Comments 4: Figure 4 and b should be collected in one figure as in figure 5. |
||
|
Response 4: Thank you for pointing this out. We agree with this comment. Therefore, We have combined a and b in Figure 4 into one. |
||
|
Comments 5: Figure 6 is not clear please make the contrast is more good. |
||
|
Response 5: Thank you for pointing this out. We apologized that the clarity of the image has affected you, we have replaced all images in the text with clearer ones wherever possible. |
||
|
Comments 6: Citation is very low, please ad more citation and discuss your results based on the others results. |
||
|
Response 6: Thank you for pointing this out. We have added additions and discussion of the results in the text, increasing citations
|
||
|
Comments 7: Conclusion is too long and should be shorten to half. |
||
|
Response 7: Thank you for pointing this out. We have modified the conclusion section |
||

Reviewer 2 Report
Comments and Suggestions for Authors
Thank you very much article submission in Biology journal. Article is interesting but there are a lot of improvements are required.
1. The article is written with very poor English. English language must be improved.
2. The reference style need a check.
3. What is the content of total saponins after the extraction using eutectic solvents.??
4. How much this method is more appropriate than traditional methods.??Explain with references.
5. There is a poor quality of figures and tables. Please redraw.
6. Rewrite the abstract and conclusion section.
Comments on the Quality of English Language
The English language must be improved.
Author Response
Response 6:
Thank you for pointing this out. We agree with this comment. We have reworked and rewritten the abstract and conclusion sections

Reviewer 3 Report
Comments and Suggestions for Authors
In this study, a highly efficient and environmentally friendly extraction method for tea saponin from Camellia oleifera Abel pomace, utilizing a deep eutectic solvent (DES) and ultrasonic extraction in conjunction with cellulase to obtain tea saponins, was established and optimized to a binary DES composed of betaine and ethylene glycol (mole ratio of 1:3). Optimal conditions for ultrasonic extraction, including ultrasound time, ultrasound power, and material-to-liquid ratio gave a tea saponin extraction rate of 20.93±0.48%.
The major issue with this manuscript relates to the methods used to extract and purify tea saponin or the lack of detail thereof. The title and introduction indicate that a new extraction procedure involving using deep eutectic solvents (DECs) is described. However, the details of preparing DECs [ln 109-115] and subsequent extraction procedures [ln 156-166] are vague, confusing, and uninformative, to say the least.
While the bioactivities of the new tea saponin preparation are thoroughly analyzed and described, the text does not provide a single detailed, encompassing summation of the optimum procedure. This information is a fundamental requirement for researchers to reproduce the study.
Ln 43 ‘resistantce’. Resistance?
Ln 77 ‘Throughout the experiment’. Delete this text.
Ln 143 ‘10 ml of water-containing DES’. 10 ml of water/ DES (7:3 v/v)?
Ln 184 Specify the cell types used.
-------------------------------------------------------------------------------
Ln 156-166 2.7. Purification of tea saponins from Camellia oleifera Abel seed pomace.
[description given in manuscript]
The methodology employed by Tang22 was utilized for the isolation and purification of tea saponins using macroporous resin. Before adsorption with D101 macroporous resin, the resin underwent pre-soaking in 95% ethanol, 5% HCl, and 5% NaOH for 12 hours, with subsequent neutralization after each soaking. 3 times the volume of 90% ethanol was added to the extracted solvent at 75℃ for 3 hours. The resulting precipitate was removed, and the supernatant was collected and concentrated using a rotary evaporator. The concentrated solution was then subjected to adsorption using the pre-treated resin, which was subsequently eluted with deionized water and 80% ethanol. The eluate was collected, concentrated using a rotary evaporator, and subsequently lyophilized in a freezing dryer. The lyophilized tea saponin samples were used in subsequent tests.
Comments on the Quality of English LanguageMinor editing required.
Author Response
|
1. Summary |
|
|
|
Thank you very much for taking the time to review this manuscript. Please find the detailed responses below and the corresponding revisions/corrections highlighted/in track changes in the re-submitted files. |
||
|
3. Point-by-point response to Comments and Suggestions for Authors |
||
|
Comments 1: The major issue with this manuscript relates to the methods used to extract and purify tea saponin or the lack of detail thereof. The title and introduction indicate that a new extraction procedure involving using deep eutectic solvents (DECs) is described. However, the details of preparing DECs [ln 109-115] and subsequent extraction procedures [ln 156-166] are vague, confusing, and uninformative, to say the least.While the bioactivities of the new tea saponin preparation are thoroughly analyzed and described, the text does not provide a single detailed, encompassing summation of the optimum procedure. This information is a fundamental requirement for researchers to reproduce the study. |
||
|
Response 1: Thank you for pointing this out. We have added descriptions to the section of the article on extraction and purification, and we hope that the revised article will compound the requirements. |
||
|
Comments 2: Ln 43 ‘resistantce’. Resistance? |
||
|
Response 2: Sorry for this type of problem in the article, we have made a change in the article |
||
|
Comments 3: Ln 77 ‘Throughout the experiment’. Delete this text. |
||
|
Response 3: Thank you for pointing this out. We agree with this comment. We have removed this text |
||
|
Comments 4: Ln 143 ‘10 ml of water-containing DES’. 10 ml of water/ DES (7:3 v/v)? |
||
|
Response 4: Thank you for pointing this out. We agree with this comment. Therefore, We have added explanations and clarifications in the text |
||
|
Comments 5: Ln 184 Specify the cell types used. |
||
|
Response 5: Thank you for pointing this out. We have amended the reference formatting in the text as required
|
||

Reviewer 4 Report
Comments and Suggestions for Authors
The manuscript "The extraction and evaluation of tea saponin using deep eutectic solvents" provides interesting data on the use of deep eutectic solvents for the extraction of biologically active substances. The use of deep eutectic solvents for the extraction of biologically active substances is undoubtedly a current scientific direction. However, there are some comments about the manuscript:
- The abstract should provide more specific results obtained.
- Authors should add keywords for better recognition of the article.
- The text of the manuscript should be checked carefully. There are typos (punctuation, extra spaces, signs, etc.).
- In the introduction, it is worth considering in more detail the available options for extracting tea saponin (advantages, disadvantages). Provide examples of articles on the use of deep eutectic solvents for the extraction of tea saponin from Camellia oleifera Abel.
- Table 1 – HBA and HBD? please decrypt
- Why was the enzyme cellulase added? At what stage and under what conditions was the enzyme added?
- More detailed information about the equipment used (brand, manufacturer, etc.) should be added to the “Materials and Methods” section.
- The section "Materials and Methods" should provide more detailed information about the experimental conditions (temperature, centrifugation conditions, extraction conditions, concentration, etc.).
- L 125-126: “The obtained 125 data can be plotted on a graph.” I think this proposal is unnecessary here.
- Paragraph 2.5 is not clear what this stage of research is. What are the conditions?
- Paragraph 2.8.2 what cell cultures were used?
- L 184-186: It is not clear what substance was tested. The name of the test substance should be indicated. Add a link to previously conducted research.
- What test strains were used in paragraph 2.8.3?
- Describe the Oxford cup method in more detail.
- In the section "Materials and Methods" you should add the item "Statistics".
- It is worth adding more detailed information on the results of the synthesis of deep eutectic solvents. How was the success of the synthesis assessed? Have any studies been carried out on rheological properties and thermal properties?
- Why was the 3:7 ratio between DES and water used? In general, it is not entirely clear why the authors choose DES Bet-EG? According to Figure 3, not the smallest amount of tea saponin is extracted simply with water.
- The text of the manuscript must provide references to relevant additional materials. For example, without appropriate references, it is not clear on what basis the authors draw their conclusions in paragraph 3.3.
- L 291-298: It seems to me that this information should be presented in the “Materials and Methods” section.
- Is it possible to combine Figures 4a and 4b? What is the degree of purity according to the manufacturer?
- Are the results of the antioxidant activity of extracted tea saponins consistent with the results of other researchers?
- Please add a method for studying cell morphology to the “Materials and Methods” section.
- Figure 7 needs to be improved.
- It is unclear from the text what concentrations of tea saponin were used in the analysis of antibacterial activity in paragraph 3.7.
- Please check the results presented in Table 3. (S. pneumoniae R 12.037±0.22?)
- Supplementary materials Figure 4. The figures must indicate the concentrations being studied.
Comments on the Quality of English LanguageThe text of the manuscript should be checked carefully. There are typos (punctuation, extra spaces, signs, etc.).
Author Response
|
1. Summary |
|
|
|
Thank you very much for taking the time to review this manuscript. Please find the detailed responses below and the corresponding revisions/corrections highlighted/in track changes in the re-submitted files. |
||
|
3. Point-by-point response to Comments and Suggestions for Authors |
||
|
Comments 1: The abstract should provide more specific results obtained.se of deep eutectic solvents for the extraction of tea saponin from Camellia oleifera Abel. |
||
|
Response 1: Thank you for your comments, we have revised the abstract section of the article. and added a description of the tea saponins in the introduction section. We have revised the full text as much as possible for typos and other issues. We hope that we can meet the requirements. |
||
|
Comments 2: Authors should add keywords for better recognition of the article. |
||
|
Response 2: Thank you for your suggestion, we will change the keyword section. |
||
|
Comments 3: The text of the manuscript should be checked carefully. There are typos (punctuation, extra spaces, signs, etc.). |
||
|
Response 3: Thanks to your suggestion, we have made a complete correction to the typo problem in the text |
||
|
Comments 4: In the introduction, it is worth considering in more detail the available options for extracting tea saponin (advantages, disadvantages). Provide examples of articles on the use of deep eutectic solvents for the extraction of tea saponin from Camellia oleifera Abel. |
||
|
Response 4: Thank you for pointing this out, we have added this section to the text. |
||
|
Comments 5: Table 1 – HBA and HBD? please decrypt |
||
|
Response 5: Thanks to your suggestion, we've added explanations both in the text and in the table |
||
|
Comments 6: Why was the enzyme cellulase added? At what stage and under what conditions was the enzyme added? |
||
|
Response 6: Thank you for your question. Cellulase was added to further increase the extraction rate. Cellulase added to the extraction fraction. And we've added a note in the text. |
||
|
Comments 7: More detailed information about the equipment used (brand, manufacturer, etc.) should be added to the “Materials and Methods” section. |
||
|
Response 7: Thanks to your suggestion, we have added instructions in the Materials and Instruments section. |
||
|
Comments 8: The section "Materials and Methods" should provide more detailed information about the experimental conditions (temperature, centrifugation conditions, extraction conditions, concentration, etc.). |
||
|
Response 8: Thanks to your suggestion, we have added experimental conditions in the text |
||
|
Comments 9: L 125-126: “The obtained 125 data can be plotted on a graph.” I think this proposal is unnecessary here. |
||
|
Response 9: Thank you for your suggestion, this part has been removed. |
||
|
Comments 10: Paragraph 2.5 is not clear what this stage of research is. What are the conditions? |
||
|
Response 10: Thanks to your suggestion, we have added this section to the text |
||
|
Comments 11: Paragraph 2.8.2 what cell cultures were used? |
||
|
Response 11: Thank you for your suggestion and we have added a description of the cultures used in the materials section |
||
|
Comments 12: L 184-186: It is not clear what substance was tested. The name of the test substance should be indicated. Add a link to previously conducted research. |
||
|
Response 12: Thanks to your suggestion, we've added additional descriptions to this section |
||
|
Comments 13: What test strains were used in paragraph 2.8.3? |
||
|
Response 13: Thanks to your suggestion, we have added a description of the strains used in this section |
||
|
Comments 14: Describe the Oxford cup method in more detail. |
||
|
Response 14: Thanks for the suggestion, have added a description to the section |
||
|
Comments 15: In the section "Materials and Methods" you should add the item "Statistics". |
||
|
Response 15: Thank you for your suggestion, the statistics section has been added |
||
|
Comments 16: It is worth adding more detailed information on the results of the synthesis of deep eutectic solvents. How was the success of the synthesis assessed? Have any studies been carried out on rheological properties and thermal properties? |
||
|
Response 16: Thanks to your suggestion, we evaluated the successful synthesis of the solvents by the low eutectic solvents that did not precipitate crystals at room temperature and maintained a stable flow. we aren’t carried out on rheological properties and thermal properties |
||
|
Comments 17: Why was the 3:7 ratio between DES and water used? In general, it is not entirely clear why the authors choose DES Bet-EG? According to Figure 3, not the smallest amount of tea saponin is extracted simply with water. |
||
|
Response 17: According to previous studies, the enzyme works better in aqueous low eutectic solvents, and the ratio of low eutectic solvents to water was determined to be 3:7 in consideration of the experimental manipulation. |
||
|
Comments 18: The text of the manuscript must provide references to relevant additional materials. For example, without appropriate references, it is not clear on what basis the authors draw their conclusions in paragraph 3.3. |
||
|
Response 18: Thanks for the suggestion, it has been added to the article |
||
|
Comments 19: L 291-298: It seems to me that this information should be presented in the “Materials and Methods” section. |
||
|
Response 19: Thanks for the suggestion, it has been changed in the article |
||
|
Comments 20: Is it possible to combine Figures 4a and 4b? What is the degree of purity according to the manufacturer? |
||
|
Response 20: Thanks for the suggestion, have merged the images and added the question about the concentration of the standards |
||
|
Comments 21: Are the results of the antioxidant activity of extracted tea saponins consistent with the results of other researchers? |
||
|
Response 21: Thank you for your suggestion, discussion of findings with others has been added to the corresponding section |
||
|
Comments 22: Please add a method for studying cell morphology to the “Materials and Methods” section. |
||
|
Response 22: Thanks for the suggestion, have added this section |
||
|
Comments 23: Figure 7 needs to be improved. |
||
|
Response 23: Thanks for the suggestion, have increased the clarity of the images as much as possible! |
||
|
Comments 24: It is unclear from the text what concentrations of tea saponin were used in the analysis of antibacterial activity in paragraph 3.7. |
||
|
Response 24: Thank you for your suggestion, a description of the concentration of tea saponin has been added. |
||
|
Comments 25: Please check the results presented in Table 3. (S. pneumoniae R 12.037±0.22?) |
||
|
Response 25: Thank you for your suggestion, the corresponding part has been adjusted |
||
|
Comments 26: Supplementary materials Figure 4. The figures must indicate the concentrations being studied. |
||
|
Response 26: Thank you for your suggestions, additional concentrations have been added |
||

Reviewer 5 Report
Comments and Suggestions for Authors In the current study, Guo and colleagues developed a novel method for the extraction of tea saponin from Camellia oleifera Abel pomace using a deep eutectic solvent approach based on betain and ethylene glycol (1:3). In addition to identifying the optimal extraction scheme, the authors evaluated the antioxidant potency of tea saponin, its cytotoxic effect against cancer cells, its effects on SOD activity and ROS levels, as well as its antibacterial potency on a panel of clinically relevant bacterial strains. This work deserves attention, is characterized by scientific novelty, and can be published in Biology after the following comments are fixed. Major comments: 1) The authors need to confirm the structure of the isolated tea saponin by NMR analysis. This would greatly improve the present work. 2) Section 3.3 - please illustrate your extraction optimization with the obtained results in the form of tables, graphs, or heat map. This is to better understand why the A3B2C1 scheme (line 287) was chosen. 3) Please perform a statistical analysis of the obtained data and mark statistically significant differences from the control on the graphs and tables. 4) Section 3.6 - Please evaluate the cytotoxicity of the investigated tea saponins using the standard MTT assay or its analogues (to make assumptions about dose-dependent changes in SOD and ROS, it is necessary to correlate these changes with the number of living cells; bright field microscopy is not sufficient in this case). 5) Figure 5 - Please reformat the graphs: (1) enlarge/replace/color markers to better separate control from experimental group; (2) increase font size of legend. 6) Figure 7 needs serious corrections. In the graphs, it is very difficult to determine which group is which and how much of the observed differences are statistically significant. Please correct this. For example, you could use different colors to color the curves and increase the size of the markers. Minor comments: line 99 - insert a space between cellulase and (10000 U/g). lines 103 - 105 - names of bacterial species should be italicized. lines 111, 112 - please provide an explanation of the abbreviations HBA and HBD Table 1 - please delete the dot after Mole ratio line 136 - please remove the extra space between wavelength and at and insert a space between 10 and μl. line 140 - please insert a space between 0.45 and μm. line 143 - please insert a space between 0.1 and g line 152 - move comma to selected lines 178-180, 186, 187, 296, 298, 313 - please insert missing spaces between numbers and units (e.g. 0.4 mg/ml). line 199 - please correct to mg/mL. Please decide how you will denote liters by l or L and make everything look the same throughout the manuscript. line 210 - please insert a space between 100 and μl. Do not use u instead of μ. line 220 - please correct to DESs (Table 2). line 270 - please move comma to saponin line 299 - please remove the extra dot and space Figure 4 - please give the names of the OX and OY axes. What is AU? Figure 6 - please perform a statistical analysis to determine if the observed differences between groups are reliable. line 381 - correct the font at the beginning of the sentence. Table 3 - please indicate what the letters a, b, c, d stand for. line 410 - please remove the extra space between activity and dot. line 412 - please write Camellia oleifera in italicsAuthor Response
|
1. Summary |
|
|
|
Thank you very much for taking the time to review this manuscript. Please find the detailed responses below and the corresponding revisions/corrections highlighted/in track changes in the re-submitted files. |
||
|
2. Point-by-point response to Comments and Suggestions for Authors |
||
|
Comments 1: The authors need to confirm the structure of the isolated tea saponin by NMR analysis. This would greatly improve the present work. |
||
|
Response 1: Thank you very much for your suggestion, but I'm afraid this experiment cannot be done at this time due to laboratory constraints |
||
|
Comments 2: Section 3.3 - please illustrate your extraction optimization with the obtained results in the form of tables, graphs, or heat map. This is to better understand why the A3B2C1 scheme (line 287) was chosen. |
||
|
Response 2: Thank you for your suggestion, regarding the results of the extraction optimisation, we have represented them in tabular form in the attached table. And we have added instructions in the article |
||
|
Comments 3: Please perform a statistical analysis of the obtained data and mark statistically significant differences from the control on the graphs and tables. |
||
|
Response 3: Thank you for your suggestion, the form of the significance analysis has been modified |
||
|
Comments 4: Section 3.6 - Please evaluate the cytotoxicity of the investigated tea saponins using the standard MTT assay or its analogues (to make assumptions about dose-dependent changes in SOD and ROS, it is necessary to correlate these changes with the number of living cells; bright field microscopy is not sufficient in this case). |
||
|
Response 4: Thank you for your suggestion, in this study, only the changes in SOD and ROS before and after the treatment of the cells with tea saponin were taken into account, and the number of live cells did not have a significant impact on the experiments |
||
|
Comments 5: Figure 5 - Please reformat the graphs: (1) enlarge/replace/color markers to better separate control from experimental group; (2) increase font size of legend. |
|
Response 5 Thank you for your suggestion, the chart in the text has been corrected |
|
Comments 6: Figure 7 needs serious corrections. In the graphs, it is very difficult to determine which group is which and how much of the observed differences are statistically significant. Please correct this. For example, you could use different colors to color the curves and increase the size of the markers. |
|
Response 6: Thank you for your suggestion, the chart in the text has been corrected |
|
Comments 7: Minor comments: line 99 - insert a space between cellulase and (10000 U/g). lines 103 - 105 - names of bacterial species should be italicized. lines 111, 112 - please provide an explanation of the abbreviations HBA and HBD Table 1 - please delete the dot after Mole ratio line 136 - please remove the extra space between wavelength and at and insert a space between 10 and μl. line 140 - please insert a space between 0.45 and μm. line 143 - please insert a space between 0.1 and g line 152 - move comma to selected lines 178-180, 186, 187, 296, 298, 313 - please insert missing spaces between numbers and units (e.g. 0.4 mg/ml). line 199 - please correct to mg/mL. Please decide how you will denote liters by l or L and make everything look the same throughout the manuscript. line 210 - please insert a space between 100 and μl. Do not use u instead of μ. line 220 - please correct to DESs (Table 2). line 270 - please move comma to saponin line 299 - please remove the extra dot and space Figure 4 - please give the names of the OX and OY axes. What is AU? Figure 6 - please perform a statistical analysis to determine if the observed differences between groups are reliable. line 381 - correct the font at the beginning of the sentence. Table 3 - please indicate what the letters a, b, c, d stand for. line 410 - please remove the extra space between activity and dot. line 412 - please write Camellia oleifera in italic |
|
Response 7: Thank you for your suggestions, we have made changes to the minor issues you raised. and did a check on the article as a whole. |

Round 2
Reviewer 3 Report
Comments and Suggestions for Authors
The issues raised in the review have been addressed in a satisfactory manner.
Comments on the Quality of English LanguageThe English can still be improved.
Reviewer 4 Report
Comments and Suggestions for Authors
After re-reviewing, I have no comments.
Reviewer 5 Report
Comments and Suggestions for Authors
The authors have successfully corrected the text according to the previously identified deficiencies. The manuscript is now ready to be published. I wish the authors success in their further research!